# Win: Weight-Decay-Integrated Nesterov Acceleration for Adaptive Gradient Algorithms

**Pan Zhou**[1]    **Xingyu Xie**[1,2]    **Shuicheng Yan**[1]
[1]Sea AI Lab        [2]Peking University
{zhoupan,xyxie,yansc}@sea.com        xyxie@pku.cn

## Abstract

Training deep networks on increasingly large-scale datasets is computationally challenging. In this work, we explore the problem of "*how to accelerate the convergence of adaptive gradient algorithms in a general manner*", and aim at providing practical insights to boost the training efficiency. To this end, we propose an effective Weight-decay-Integrated Nesterov acceleration (Win) for adaptive algorithms to enhance their convergence speed. Taking AdamW and Adam as examples, we minimize a dynamical loss per iteration which combines the vanilla training loss and a dynamic regularizer inspired by proximal point method (PPM) to improve the convexity of the problem. Then we respectively use the first- and second-order Taylor approximations of vanilla loss to update the variable twice while fixing the above dynamic regularization brought by PPM. In this way, we arrive at our Win acceleration (like Nesterov acceleration) for AdamW and Adam that uses a conservative step and a reckless step to update twice and then linearly combines these two updates for acceleration. Next, we extend this Win acceleration to LAMB and SGD. Our transparent acceleration derivation could provide insights for other accelerated methods and their integration into adaptive algorithms. Besides, we prove the convergence of Win-accelerated adaptive algorithms by taking AdamW and Adam as examples. Experimental results testify the faster convergence speed and superior performance of our Win-accelerated AdamW, Adam, LAMB and SGD over their vanilla counterparts on vision classification tasks and language modeling tasks with CNN and Transformer backbones.

## 1 Introduction

Deep neural networks (DNNs) are effective to model realistic data and have been successfully applied to various applications, *e.g.* image classification [1–10] and speech recognition [11–14]. Typically, their training models can be formulated as the following nonconvex optimization problem:

$$\min_{\boldsymbol{z}\in\mathbb{R}^d} F(\boldsymbol{z}) := \mathbb{E}_{\boldsymbol{\zeta}\sim\mathcal{D}}[f(\boldsymbol{z},\boldsymbol{\zeta})] + \frac{\lambda}{2}\|\boldsymbol{z}\|_2^2, \tag{1}$$

where $\boldsymbol{z}$ is the model parameters; sample $\boldsymbol{\zeta}$ is drawn from a data distribution $\mathcal{D}$; the loss $f$ is differentiable; $\lambda$ is a constant. To solve problem (1), SGD [15, 16] uses its compositional structure to efficiently estimate gradient via minibatch data, and has become a dominant algorithm to train DNNs. However, on sparse data or ill-conditioned problems, SGD suffers from slow convergence speed [17, 18], as it scales the gradient uniformly in all parameter coordinate and ignores the data or problem properties on each coordinate. To resolve this issue, recent work has proposed a variety of adaptive methods, *e.g.* Adam [17] and AdamW [19], that scale each gradient coordinate according to the current geometry curvature of the loss $F(\boldsymbol{z})$. This coordinate-wise scaling greatly accelerates the optimization convergence and helps them, *e.g.* Adam and AdamW, become much more popular in DNN training.

Has it Trained Yet? Workshop at the Conference on Neural Information Processing Systems (NeurIPS 2022).

Unfortunately, along with the increasing scale of both datasets and models, efficient DNN training even with SGD or adaptive algorithms has become very challenging. In this work, we are particularly interested in the problem of "*how to accelerate the convergence of adaptive algorithms in a general manner*" because of their dominant popularity across many DNNs. Heavy ball acceleration [20] and Nesterov acceleration [21] are widely used in SGD but are rarely studied in adaptive algorithms.

**Contributions:** In this work, based on a recent Nesterov-type acceleration formulation [22] and proximal point method (PPM) [23], we propose a new *Weight-decay-Integrated Nesterov acceleration* (Win) to accelerate adaptive algorithms. By taking AdamW and Adam as examples, we follow PPM spirit and minimize a dynamically regularized loss which combines vanillas loss and a dynamical regularization, and independently approximate the vanilla loss by its first- and second-order Taylor expansions to update the variable twice while fixing the above dynamic regularization. As a result, we achieve at our Win acceleration, a Nesterov-alike acceleration, for AdamW and Adam that uses a conservative step and a reckless step to update twice and then linearly combines these two updates for acceleration. Then we extend Win acceleration to LAMB [24] and SGD. This transparent acceleration derivation may motivate other accelerations and provide examples to introduce other accelerations into adaptive algorithms. Moreover, we analyze the convergence of Win-accelerated adaptive algorithms to justify their convergence superiority by using AdamW & Adam as examples.

Finally, experimental results on both vision classification tasks and language modeling tasks show that our Win-accelerated algorithms, i.e. accelerated AdamW, Adam, LAMB and SGD, can accelerate the convergence speed and also improve the performance of their corresponding non-accelerated counterparts by a remarkable margin on both CNN and transformer architectures.

## 2  Weight-decay-Integrated Nesterov Acceleration

In deterministic optimization, one widely used optimization-stabilizing and acceleration approach is proximal point method (PPM) [23, 25]. At the $k$-th iteration, PPM optimizes an $\ell_2$-regularized loss $F(\boldsymbol{z}) + \frac{1}{2\eta_k}\|\boldsymbol{z} - \boldsymbol{z}_{k-1}\|_2^2$ instead of the vanilla loss $F(\boldsymbol{z})$. This change enhances the convexity of the problem, accelerating and also stabilizing optimization [26, 27]. To make the $\ell_2$-regularized problem solvable iteratively, PPM approximates $F(\boldsymbol{z})$ by its first- or second-order Taylor expansion to get a close-form solution. At below, we borrow the idea in PPM to induce a *Weight-decay-Integrated Nesterov acceleration* (Win) for adaptive algorithms by using AdamW and Adam as examples.

**Win-Accelerated AdamW and Adam.** To begin with, following most adaptive gradient algorithms, *e.g.* Adam and AdamW, we estimate the first- and second-order moments $\boldsymbol{m}_k$ and $\boldsymbol{v}_k$ of gradient as

$$\boldsymbol{g}_k = \frac{1}{b}\sum\nolimits_{i=1}^{b}\nabla f(\boldsymbol{z}_k; \boldsymbol{\zeta}_i), \quad \boldsymbol{m}_k = (1-\beta_1)\boldsymbol{m}_{k-1} + \beta_1 \boldsymbol{g}_k, \quad \boldsymbol{v}_k = (1-\beta_2)\boldsymbol{v}_{k-1} + \beta_2 \boldsymbol{g}_k^2, \quad (2)$$

where $\boldsymbol{m}_0 = \boldsymbol{g}_0$, $\boldsymbol{v}_0 = \boldsymbol{g}_0^2$, $\beta_1 \in [0, 1]$ and $\beta_2 \in [0, 1]$. For brevity, with a small scaler $\nu > 0$, we define

$$\boldsymbol{s}_k = \sqrt{\boldsymbol{v}_k + \nu}, \qquad \boldsymbol{u}_k = \boldsymbol{m}_k / \sqrt{\boldsymbol{v}_k + \nu}. \quad (3)$$

Then following PPM spirit, at the $k$-th iteration, we minimize a regularized loss $F(\boldsymbol{x}) + \frac{1}{2\eta}\|\boldsymbol{x} - \boldsymbol{x}_k\|_{\boldsymbol{s}_k}^2$. Here we use the regularizer $\|\boldsymbol{x} - \boldsymbol{x}_k\|_{\boldsymbol{s}_k}^2$ instead of the $\ell_2$-regularization $\|\boldsymbol{x} - \boldsymbol{x}_k\|_2^2$, since 1) this new regularization allows us to handle adaptive algorithms as shown below Eqn. (4), and 2) it also helps increase the problem convexity to speed up the convergence. To make problem solvable iteratively, we approximate $F(\boldsymbol{z})$ by its first-order Taylor expansion at the point $\boldsymbol{z}_k$ and update $\boldsymbol{x}_{k+1}$ as

$$\boldsymbol{x}_{k+1} = \operatorname{argmin}_{\boldsymbol{x}} F(\boldsymbol{z}_k) + \langle \boldsymbol{m}_k, \boldsymbol{x} - \boldsymbol{z}_k\rangle + \frac{1}{2\eta_k}\|\boldsymbol{x} - \boldsymbol{x}_t\|_{\boldsymbol{s}_k}^2 + \frac{\lambda}{2}\|\boldsymbol{x}\|_{\boldsymbol{s}_k}^2 = \frac{1}{1+\lambda\eta_k}(\boldsymbol{x}_k - \eta_k \boldsymbol{u}_k), \quad (4)$$

where $\|\boldsymbol{x}\|_{\boldsymbol{s}_k} = \sqrt{\langle \boldsymbol{x}, \boldsymbol{s}_k * \boldsymbol{x}\rangle}$ with element-wise product $*$, $\boldsymbol{m}_k$ is used to approximate the full gradient $\nabla F(\boldsymbol{z}_k)$ for Taylor expansion. We add a small regularization $\frac{\lambda}{2}\|\boldsymbol{x}\|_{\boldsymbol{s}_k}^2$, as 1) it can improve the generalization in practice [19, 28]; 2) it allows us to derive Adam ($\lambda = 0$) and AdamW ($\lambda > 0$). If $\lambda = 0$, the updating (4) becomes the exact Adam algorithm. If $\lambda > 0$, the updating (4) can approximate the updating rule $\boldsymbol{x}_{k+1} = (1 - \lambda\eta_k)\boldsymbol{x}_k - \eta_k \boldsymbol{u}_k$ of AdamW. This is because consider $\lambda\eta_k$ is small in practice, we approximate $(1 + \lambda\eta_k)^{-1} = 1 - \lambda\eta_k + \mathcal{O}(\lambda^2\eta_k^2)$ and thus $\frac{1}{1+\lambda\eta_k}(\boldsymbol{x}_k - \eta_k \boldsymbol{u}_k) = [1 - \lambda\eta_k + \mathcal{O}(\lambda^2\eta_k^2)]\boldsymbol{x}_k - [\eta_k - \mathcal{O}(\lambda\eta_k^2) + \mathcal{O}(\lambda^3\eta_k^3)]\boldsymbol{u}_k$ which becomes AdamW

---

**Algorithm 1: Win-Accelerated AdamW, Adam and LAMB**

**Input:** initialization $\boldsymbol{x}_0 = \boldsymbol{z}_0 = \boldsymbol{0}$, step size $\{(\eta_k, \bar{\eta}_k)\}_{k=0}^T$, moment parameters $\{\beta_1, \beta_2\}$.

1 **while** $k < T$ **do**

2     $\boldsymbol{g}_k = \frac{1}{b} \sum_{i=1}^b \nabla f(\boldsymbol{z}_k; \zeta_i)$

3     $\boldsymbol{m}_k = (1 - \beta_1)\boldsymbol{m}_{k-1} + \beta_1 \boldsymbol{g}_k$                         /* $\boldsymbol{m}_0 = \boldsymbol{g}_0$ */

4     $\boldsymbol{v}_k = (1 - \beta_2)\boldsymbol{v}_{k-1} + \beta_2 \boldsymbol{g}_k^2$                         /* $\boldsymbol{v}_0 = \boldsymbol{g}_0^2$ */

5     $\boldsymbol{u}_k = \frac{\boldsymbol{m}_k}{\sqrt{\boldsymbol{v}_k} + \nu}$ for AdamW and Adam, $\boldsymbol{u}_k = \frac{\|\boldsymbol{x}_k\|_2}{\|\boldsymbol{m}_k / \sqrt{\boldsymbol{v}_k} + \nu\|_2} \frac{\boldsymbol{m}_k}{\sqrt{\boldsymbol{v}_k} + \nu}$ for LAMB

6     $\boldsymbol{x}_{k+1} = \frac{1}{1 + \lambda \eta} (\boldsymbol{x}_k - \eta_k \boldsymbol{u}_k)$

7     $\boldsymbol{z}_{k+1} = \bar{\eta}_k \tau_k \boldsymbol{x}_{k+1} + \eta_k \tau_k (\boldsymbol{z}_k - \bar{\eta}_k \boldsymbol{u}_k)$ with $\tau_k = \frac{1}{\eta_k + \bar{\eta}_k + \lambda \eta_k \bar{\eta}_k}$

8 **end while**

---

by ignoring $\mathcal{O}(\eta_k^2)$ and $\mathcal{O}(\eta_k^3)$. This is one reason that we adopt the regularizer $\|\boldsymbol{x} - \boldsymbol{x}_k\|_{\boldsymbol{s}_k}^2$ in (4) instead of the $\ell_2$-regularization in PPM, as we can flexibly derive Adam and AdamW.

Similarly, we minimize a regularized loss $F(\boldsymbol{z}) + \frac{1}{2\eta_k}\|\boldsymbol{z} - \boldsymbol{x}_{t+1}\|_{\boldsymbol{s}_k}^2$, and further approximate $F(\boldsymbol{z})$ by its second-order approximation $F(\boldsymbol{z}_k) + \langle \boldsymbol{m}_k, \boldsymbol{z} - \boldsymbol{z}_k \rangle + \frac{1}{2\bar{\eta}_k}\|\boldsymbol{z} - \boldsymbol{z}_k\|_{\boldsymbol{s}_k}^2$:

$$\boldsymbol{z}_{k+1} = \arg\min_{\boldsymbol{z}} F(\boldsymbol{z}_k) + \langle \boldsymbol{m}_k, \boldsymbol{z} - \boldsymbol{z}_k \rangle + \frac{1}{2\bar{\eta}_k}\|\boldsymbol{z} - \boldsymbol{z}_k\|_{\boldsymbol{s}_k}^2 + \frac{1}{2\eta_k}\|\boldsymbol{z} - \boldsymbol{x}_{k+1}\|_{\boldsymbol{s}_k}^2 + \frac{\lambda}{2}\|\boldsymbol{z}\|_{\boldsymbol{s}_k}^2$$
$$= \bar{\eta}_k \tau_k \boldsymbol{x}_{k+1} + \eta_k \tau_k (\boldsymbol{z}_k - \bar{\eta}_k \boldsymbol{u}_k), \tag{5}$$

where $\tau_k = \frac{1}{\eta_k + \bar{\eta}_k + \lambda \eta_k \bar{\eta}_k}$, $\boldsymbol{m}_k$ can approximate $\nabla F(\boldsymbol{x}_k)$ as shown in Theorem 1 in Appendix B.

For more clear, we introduce a virtual sequence $\boldsymbol{y}_{k+1} = \boldsymbol{z}_k - \bar{\eta}_k \boldsymbol{u}_k$ in Win, and rewrite (5) as

$$\boldsymbol{x}_{k+1} = (1 + \lambda \eta_k)^{-1}(\boldsymbol{x}_k - \eta_k \boldsymbol{u}_k), \quad \boldsymbol{y}_{k+1} = \boldsymbol{z}_k - \bar{\eta}_k \boldsymbol{u}_k, \quad \boldsymbol{z}_{k+1} = \bar{\eta}_k \tau_k \boldsymbol{x}_{k+1} + \eta_k \tau_k \boldsymbol{y}_{k+1}. \tag{6}$$

See detailed steps in Algorithm 1. Interestingly, Win acceleration is similar to Nesterov-type acceleration, since they both use a conservative step $\eta_k$ and a reckless step $\bar{\eta}_k$ to update $\boldsymbol{x}_{k+1}$ and $\boldsymbol{y}_{k+1}$ respectively, and then linearly combine $\boldsymbol{x}_{k+1}$ and $\boldsymbol{y}_{k+1}$ to obtain $\boldsymbol{z}_{k+1}$.

Our Win-acceleration is quit simple and efficient, as our accelerated AdamW/Adam only adds an extra simple algorithmic step, i.e. the 7th step in Algorithm 1, on vanilla AdamW/Adam. Moreover, for the only extra hyper-parameter, the reckless step $\bar{\eta}_k$, in Algorithm 1 over AdamW/Adam, we always set it as $\bar{\eta}_k = 2\eta_k$, which works well in our all experiments.

**Extension to LAMB and SGD.** Here we generalize Win acceleration to LAMB [24] and SGD [15]. For LAMB, it scales the update $\boldsymbol{u}_k$ of AdamW in Eqn. (3) so that $\boldsymbol{u}_k$ is at the same magnitude of the network weight $\boldsymbol{x}_k$. That is, it changes the update rule $\boldsymbol{x}_{k+1} = (1 - \lambda \eta_k)\boldsymbol{x}_k - \eta_k \boldsymbol{m}_k / \boldsymbol{s}_k$ in AdamW to $\boldsymbol{x}_{k+1} = \boldsymbol{x}_k - \eta_k \frac{\|\boldsymbol{x}_k\|_2}{\|\boldsymbol{r}_k + \lambda \boldsymbol{x}_k\|_2}(\boldsymbol{r}_k + \lambda \boldsymbol{x}_k)$ where $\boldsymbol{r}_k = \boldsymbol{m}_k / \boldsymbol{s}_k$. This modification is to avoid too large or small update, improving optimization efficiency. To extend Win acceleration to LAMB, we inherit this scaling spirit, and scale the update $\boldsymbol{u}_k$ in (3) to the following one:

$$\boldsymbol{u}_k = (\|\boldsymbol{x}_k\|_2 / \|\boldsymbol{m}_k / \boldsymbol{s}_k\|_2) \cdot (\boldsymbol{m}_k / \boldsymbol{s}_k). \tag{7}$$

Next, we can follow Eqn. (4) and (5) to update, and summarize detailed steps in Algorithm 1.

For SGD, applying Win acceleration to it is quite direct. Specifically, the only algorithmic difference between SGD and AdamW on the $\ell_2$-regularized problems is that SGD has no second-order moment $\boldsymbol{v}_k$ while AdamW has. So we can borrow the acceleration framework of AdamW to accelerate SGD by setting $\boldsymbol{s}_k = \boldsymbol{1} \in \mathbb{R}^d$ in Eqn. (3), (4) and (5), and obtain WIN-accelerated SGD:

$$\boldsymbol{m}_k = \beta_1 \boldsymbol{m}_{k-1} + \beta_1' \boldsymbol{g}_k, \quad \boldsymbol{x}_{k+1} = \frac{1}{1 + \lambda \eta_k}(\boldsymbol{x}_k - \eta_k \boldsymbol{m}_k), \quad \boldsymbol{z}_{k+1} = \bar{\eta}_k \tau_k \boldsymbol{x}_{k+1} + \eta_k \tau_k (\boldsymbol{z}_k - \bar{\eta}_k \boldsymbol{m}_k), \tag{8}$$

where $\beta_1' \in [0, 1]$ is dampening parameter. Here we slightly modify the moment $\boldsymbol{m}_k$ to accord with the one used in Nesterov-accelerated SGD (*e.g.* SGD-M in Pytorch).

**Convergence Analysis.** Theorem 1 in Appendix B analyzes the convergence of Win-accelerated adaptive algorithms to justify their convergence superiority by using AdamW & Adam as examples.

Table 1: ImageNet top-1 accuracy (%) of ResNet50&101 whose official optimizer is LAMB due to the stronger data augmentation for better performance. ∗ is reported in [29].

| | ResNet50 | | | | ResNet101 | | | |
|---|---|---|---|---|---|---|---|---|
| Epoch | 100 | 200 | 300 | avg. | 100 | 200 | 300 | avg. |
| SAM | 77.3 | 78.7 | 79.4 | 78.5 | 79.5 | 81.1 | 81.6 | 80.7 |
| SGD-H | 75.3 | 76.9 | 77.2 | 76.5 | 77.7 | 78.6 | 78.8 | 78.4 |
| SGD-M | 77.0 | 78.6 | 79.3 | 78.3 | 79.3 | 81.0 | 81.4 | 80.6 |
| SGD-Win | 78.0 | 79.2 | 79.7 | $79.0_{+0.7}$ | 80.1 | 81.2 | 81.6 | $81.0_{+0.4}$ |
| Adam | 76.9 | 78.4 | 78.8 | 78.1 | 78.4 | 80.2 | 80.6 | 79.7 |
| Adam-Win | 77.8 | 78.8 | 79.3 | $78.7_{+0.6}$ | 79.2 | 80.6 | 81.0 | $80.3_{+0.6}$ |
| AdamW | 77.0 | 78.9 | 79.3 | 78.4 | 78.9 | 79.9 | 80.4 | 79.7 |
| AdamW-Win | 78.0 | 79.3 | 79.9 | $79.1_{+0.7}$ | 80.2 | 81.1 | 81.3 | $80.9_{+1.2}$ |
| LAMB | 77.0 | 79.2 | 79.8∗ | 78.7 | 79.4 | 81.1 | 81.3∗ | 80.6 |
| LAMB-Win | 78.4 | 79.7 | 80.1 | $79.4_{+0.7}$ | 80.6 | 81.5 | 81.7 | $81.3_{+0.7}$ |

Table 2: ImageNet top-1 accuracy (%) of ViT and PoolFormer whose default optimizers are both AdamW. ∗ and ⋄ are respectively reported in [28] and [30].

| | ViT-S | | | ViT-B | | | PoolFormer-S12 | | |
|---|---|---|---|---|---|---|---|---|---|
| Epoch | 150 | 300 | avg. | 150 | 300 | avg. | 150 | 300 | avg. |
| SGD-M | 77.4 | 79.4 | 78.4 | 79.6 | 80.0 | 79.8 | 69.7 | 74.3 | 72.0 |
| SGD-Win | 78.1 | 80.1 | $79.1_{+0.7}$ | 80.4 | 80.8 | $80.6_{+0.8}$ | 71.1 | 74.5 | $72.8_{+0.8}$ |
| Adam | 77.3 | 79.3 | 78.3 | 79.0 | 79.7 | 79.4 | 74.3 | 76.3 | 75.3 |
| Adam-Win | 78.6 | 80.2 | $79.4_{+1.1}$ | 80 | 80.5 | $80.3_{+0.9}$ | 75.6 | 77.1 | $76.4_{+1.1}$ |
| AdamW | 78.3 | 79.8∗ | 79.1 | 79.5 | 81.8∗ | 80.7 | 75.2 | 77.1∗ | 76.2 |
| AdamW-Win | 79.3 | 80.8 | $80.1_{+1.0}$ | 81.0 | 82.2 | $81.6_{+0.9}$ | 76.7 | 77.6 | $77.2_{+1.0}$ |
| LAMB | 78.0 | 79.6 | 78.8 | 80.3 | 80.8 | 80.6 | 75.4 | 77.4 | 76.4 |
| LAMB-Win | 79.3 | 80.6 | $80.0_{+1.2}$ | 81.0 | 81.4 | $81.2_{+0.6}$ | 76.7 | 78.0 | $77.4_{+1.0}$ |

# 3 Experiments

For clarity, we call our accelerated algorithm "X-Win", where "X" denotes vanilla optimizers. In all experiments, our accelerated algorithms, *e.g.* AdamW-Win, always use the default hyper-parameters of vanilla optimizers, *e.g.* moment parameters $\beta_1$ and $\beta_2$ in AdamW; and set $\bar{\eta}_k = 2\eta_k$.

**Results on ResNets and ViTs.** Table 1 reports accuracy of ResNets under the setting in [29], and Table 2 gives the performance of ViT [2] and PoolFormer [30]. Our accelerated algorithms always outperform their corresponding non-accelerated version. On ResNet, LAMB-Win achieves remarkable improvement over the official optimizer LAMB for this setting; SGD-Win also surpasses heavy-ball accelerated SGD (SGD-H) and Nesterov accelerated SGD (SGD-M). On ViTs, our accelerated algorithms consistently outperform the corresponding non-accelerated counterparts. Fig. 1 shows the faster faster convergence behaviors of our accelerated algorithms over non-accelerated counterparts which could benefit their better performance under the same computational cost.

**Results on Transformer-XL.** Table 3 shows that under different training steps on WikiText-103 dataset, our accelerated Adam-Win always achieves lower test PPL than the official Adam optimizer of Transformer-XL-base, and improves 1.5 average test PPL over Adam.

Table 3: Test PPL of Transformer-XL-B. ∗ is officially reported.

| Transformer-XL | Training Steps | | | |
|---|---|---|---|---|
| | 50k | 100k | 200k | avg. |
| Adam | 28.5 | 25.5 | 24.2∗ | 26.7 |
| Adam-Win | 26.7 | 25.0 | 24.0 | $25.2_{+1.5}$ |

# 4 Conclusion

In this work, we adopt proximal point method to derive a weight-decay-integrated Nesterov acceleration for AdamW and Adam, and extend it to LAMB and SGD. Moreover, we prove the convergence of our accelerated algorithms, i.e. accelerated AdamW, Adam and SGD, and observe the superiority of the accelerated Adam-type algorithm over the vanilla ones in terms of stochastic gradient complexity. Finally, experimental results validate the advantages of our accelerated algorithms.

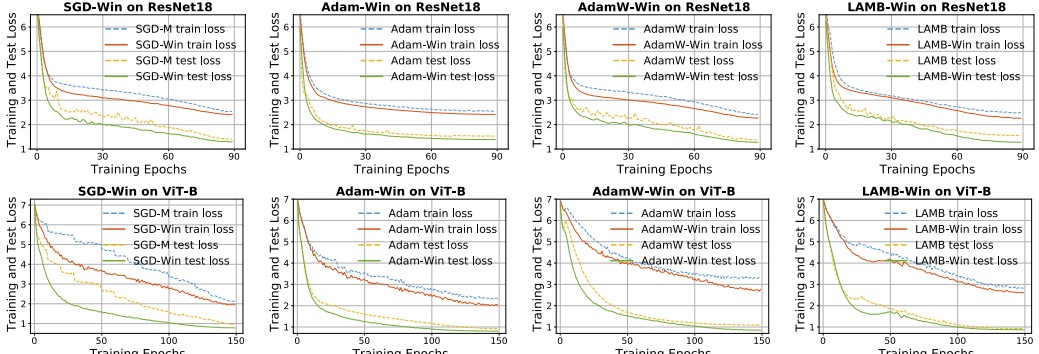

Figure 1: Training/test losses on ImageNet. Lager training loss than test one is due to its strong augmentation.

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
