# OpenReview forum: "Win: Weight-Decay-Integrated Nesterov Acceleration for Adaptive Gradient Algorithms"
_NeurIPS.cc/2022/Workshop/HITY — HITY Workshop NeurIPS 2022_

### Official Review · Reviewer_BiBe · 2022-10-12

**Rating:** 1
**Confidence:** 3

**Review:**

This paper provides an acceleration method for adaptive SGD like Adam. The core idea is to improve upon the proximal point method by taking the norm in the regularization term to be wrt. the preconditioner. The resulting method is a Nesterov-like acceleration, combining both conservative and reckless steps in a gradient update step.

I think the idea is interesting and initial experiments are looking good.

---

### Official Review · Reviewer_WJxn · 2022-10-18
**Interesting idea with good and fairly general results**

**Rating:** 1
**Confidence:** 3

**Review:**

This paper presents, in a principled way, a novel and interesting way of applying Nesterov-alike Acceleration to adaptive DL optimizers. When applied on a number of optimizers, architectures and vision classification tasks, the accelerated version shows substantial improvements across the board. The authors claim that, thanks to the principled formulation, they are able to prove advantageous convergence properties in Theorem 1, appendix B, but I wasn't able to find said appendix nor the theorem on the paper (also, some of the provided references are unused, which may be a related issue).

I think the idea of adaptive Nesterov is good and well presented, and experiments show clear advantages, so I propose to accept the paper as I consider it an interesting addition to the DL optimizer literature.

Comments
* It is true that Adam is very popular and it generally helps a lot with convergence, but it has been theoretically shown to not converge, and even fail to generalize in some cases (https://openreview.net/forum?id=ryQu7f-RZ https://arxiv.org/abs/1509.01240)
* While results look very compelling and the effort to is welcome, authors may want to consider extending to established optimizer benchmarks to enhance comparability (https://github.com/fsschneider/DeepOBS https://arxiv.org/abs/2007.01547)

---

### Decision · Program_Chairs · 2022-10-20

Accept